# Cheap Talk Discovery and Utilization in Multi-Agent Reinforcement Learning

**Yat Long Lo**
University of Oxford
Dyson Robot Learning Lab
richie.lo@dyson.com

**Christian Schroeder de Witt**
FLAIR, University of Oxford
cs@robots.ox.ac.uk

**Samuel Sokota**
Carnegie Mellon University
ssokota@andrew.cmu.edu

**Jakob Foerster**
FLAIR, University of Oxford
jakob.foerster@eng.ox.ac.uk

**Shimon Whiteson**
University of Oxford
shimon.whiteson@cs.ox.ac.uk

## Abstract

By enabling agents to communicate, recent cooperative multi-agent reinforcement learning (MARL) methods have demonstrated better task performance and more coordinated behavior. Most existing approaches facilitate inter-agent communication by allowing agents to send messages to each other through free communication channels, i.e., *cheap talk channels*. Current methods require these channels to be constantly accessible and known to the agents a priori. In this work, we lift these requirements such that the agents must discover the cheap talk channels and learn how to use them. Hence, the problem has two main parts: *cheap talk discovery* (CTD) and *cheap talk utilization* (CTU). We introduce a novel conceptual framework for both parts and develop a new algorithm based on mutual information maximization that outperforms existing algorithms in CTD/CTU settings. We also release a novel benchmark suite to stimulate future research in CTD/CTU.

## 1 Introduction

Effective communication is essential for many multi-agent systems in the partially observable setting, which is common in many real-world applications like elevator control (Crites & Barto, 1998) and sensor networks (Fox et al., 2000). Communicating the right information at the right time becomes crucial to completing tasks effectively. In the multi-agent reinforcement learning (MARL) setting, communication often occurs on free channels known as *cheap talk channels*. The agents' goal is to learn an effective communication protocol via the channel. The transmitted messages can be either discrete or continuous (Foerster et al., 2016).

Existing work often assumes the agents have prior knowledge (e.g., channel capacities and noise level) about these channels. However, such assumptions do not always hold. Even if these channels' existence can be assumed, they might not be *persistent*, i.e., available at every state. Consider the real-world application of inter-satellite laser communication. In the case, communication channel is only functional when satellites are within line of sight. This means positioning becomes essential (Lakshmi et al., 2008). Thus, Without these assumptions, agents need the capability to discover where to best communicate before learning a protocol in realistic MARL settings.

In this work, we investigate the setting where these assumptions on cheap talk channels are lifted. Precisely, these channels are only effective in a subset of the state space. Hence, agents must discover where these channels are before they can learn how to use them. We divide this problem into two sequential steps: *cheap talk discovery* (CTD) and *cheap talk utilization* (CTU). The problem is a strict generalization of the common setting used in the emergent communication literature with

less assumptions, which is more akin to real-world scenarios (see appendix A for more in-depth discussions on the setting's significance and use cases).

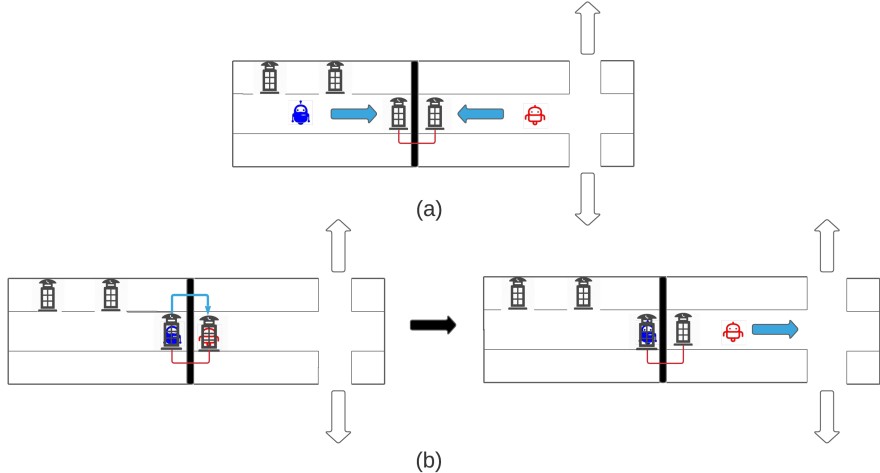

Figure 1: The two learning stages for CTD/CTU based on PBMaze. Stage (a): Discover the functional phone booths; Stage (b): Form a protocol to use the phone booth and learn to interpret the messages (left), and solve the task (right). The blue and red agents are the sender and the receiver respectively

This setting is particularly difficult as it suffers from the temporal credit assignment problem (Sutton, 1984) for communicative actions. Consider an example we call the *phone booth maze (PBMaze)*, the environment has a sender and a receiver, placed into two separate rooms. The receiver's goal is to escape from the correct exit out of two possible exits. Only the sender knows which one is the correct exit. The sender's goal is to communicate this information using functional phone booths.

This leads to two learning stages. Firstly, they need to learn to reach the booths. Then, the sender has to learn to form a protocol, distinguishing different exit information while the receiver has to learn to interpret the sender's protocol by trying different exits. This makes credit assignment particularly difficult as communicative actions do not lead to immediate rewards. Additionally, having communicative actions that are only effective in a small subset of the state space further makes it a challenging joint exploration problem, especially when communication is necessary for task completion. Figure 1 provides a visual depiction of the two learning stages in this environment.

As a whole, our contributions are four-fold. Firstly, we provide a formulation of the CTD and CTU problem. Secondly, we introduce a configurable environment to benchmark MARL algorithms on the problem. Thirdly, we propose a method to solve the CTD and CTU problems based on information theory and advances in MARL, including off-belief learning (Hu et al., 2021, OBL) and differentiable inter-agent learning (Foerster et al., 2016, DIAL). Finally, we show that our proposed approach empirically compares favourably to other MARL baselines, validate the importance of specific components via ablation studies and illustrate how our method can act as a measure of channel capacity to learn where best to communicate.

## 2 RELATED WORK

The use of mutual information (MI) has been explored in the MARL setting. Wang et al. (2019) propose a shaping reward based on MI between agents' transitions to improve exploration, encouraging visiting critical points where one can influence other agents. Our proposed method also has an MI term for reward shaping. Their measure might behave similarly to ours but is harder to compute and requires full environmental states during training. Sokota et al. (2022) propose a method to discover implicit communication protocols using environment actions via minimum entropy coupling, separating communicative and non-communicative decision-making. We propose a similar problem decomposition by separating state and action spaces into two subsets based on whether communication can occur or not. Unlike in Sokota et al. (2022), we focus on explicit communication

where specialized channels for communication exist. Jaques et al. (2019) propose rewarding agents for having causal influences over other agents' policies using MI between agents' actions. Jaques et al. (2019) is the closest to our work but still assumes the omnipresence and prior knowledge of communication channels.

Many recent papers investigate various aspects of communication in MARL. Foerster et al. (2016) propose DIAL to learn how to communicate by allowing gradients to flow across agents. We use DIAL as a component of our proposed framework. Sukhbaatar et al. (2016) propose CommNet to learn how to communicate. Unlike DIAL, it uses mean-pooling to process messages and handle a dynamic number of agents. Das et al. (2019) proposes a targeted communication architecture to tackle the issue of what messages to send and who to send them to. Singh et al. (2018) propose a gating mechanism to learn when to communicate, achieving better training efficiency scalability by reducing redundant communication. Jiang & Lu (2018) proposes an attention mechanism to learn when and who to communicate by dynamically forming communication groups. In our work, we focus on the problem of discovering where to communicate, rather than how to communicate, who to communicate, and when to communicate. Hence, our contributions are orthogonal to existing works.

## 3 BACKGROUND

Throughout this work, we consider decentralized partially observable Markov decision processes (Dec-POMDPs) with $N$ agents (Oliehoek & Amato, 2016) in the form of a tuple $G = \langle S, A, P, R, Z, \Omega, n, \gamma \rangle$. $s \in S$ is the true state of the environment. At each time step, each agent $i \in N$ chooses an action $a \in A^i$ to form a joint action $a \in A \equiv A^1 \times A^2 ... \times A^N$. This leads to a transition on the environment according to the transition function $P(s'|s, a^1, ...a^N) : S \times A \times S \to [0, 1]$. All agents share the same reward function $R(s, a) : S \times A \to \mathbb{R}$. $\gamma \in [0, 1)$ is a discount factor. Each agent $i$ receives individual observations $z \in Z$ based on the observation function $\Omega^i(s) : S \to Z$.

The environment trajectory and the action-observation history (AOH) of an agent $i$ are denoted as $\tau_t = (s_0, \mathbf{a_0}, ....s_t, \mathbf{a_t})$ and $\tau_t^i = \left(\Omega^i(s_0), a_0^i, ..., \Omega^i(s_t), a_t^i\right) \in T \equiv (Z \times A)^*$ respectively. A stochastic policy $\pi(a^i|\tau^i) : T \times A \to [0, 1]$ conditions on AOH. The joint policy $\pi$ has a corresponding action-value function $Q^\pi(s_t, a_t) = \mathbb{E}_{s_{t+1:\infty}, a_{t+1:\infty}}[R_t|s_t, a_t]$, where $R_t = \sum_{i=0}^{\infty} \gamma^i r_{t+i}$ is the discounted return. $r_{t+i}$ is the reward obtained at time $t + i$ from the reward function $R$. The distribution of states is commonly referred to as a belief $B_\pi(\tau|\tau^i) = P(\tau|\tau^i, \pi)$.

### 3.1 OFF-BELIEF LEARNING

OBL (Hu et al., 2021) is a recent method to learn policies that do not interpret the actions of other agents and assumes other agents would do the same. Precisely, it induces agents to not reason about each other's private information and actions by conditioning their beliefs only on information revealed by the environment and interpreting their actions as if they were performed by a random agent. This is often desirable as making incorrect assumptions can cause coordination failure. Therefore, learning a base policy that maximizes reward without any conventions is important, especially when agents work with others who they have not met before, a problem known as zero-shot coordination (ZSC).

The OBL operator assumes all agents to be playing a common policy $\pi_0$ up to $\tau^i$ and $\pi_1$ thereafter. Then, an agent's belief $B$, conditioned on their AOH can be computed as:

$$B_{\pi_0}(\tau|\tau^i) = P(\tau|\tau^i, \pi_0) \tag{1}$$

We denote the state-action value for playing $\pi_0$ up to $\tau^i$ and playing $\pi_1$ thereafter to be $Q^{\pi_0 \to \pi_1}(a|\tau^i)$, which is the expected return of sampling $\tau$ from $B_{\pi_0}(\tau^i)$ with all players playing $\pi_1$ starting from the end of this trajectory. The counterfactual state-action value function is defined as follows:

$$Q^{\pi_0 \to \pi_1}(a|\tau^i) = \sum_{\tau_t, \tau_{t+1}} B_{\pi_0}(\tau_t|\tau_t^i)[R(s_t, a) + \mathrm{T}(\tau_{t+1}|\tau_t)V^{\pi_1}(\tau_{t+1})]. \tag{2}$$

To compute an OBL policy using value iteration methods, the Bellman equation for $Q^{\pi_0 \to \pi_1}(\tau^i)$ for each agent $i$ is expressed as follows:

$$Q^{\pi_0 \to \pi_1}(a_t|\tau_t^i) =$$

$$\mathbb{E}_{\tau_t \sim B_{\pi_0}(\tau_t^i), \tau_{t+k} \sim (\mathrm{T}, \pi_1)} \left[ \sum_{t'=t}^{t+k-1} R(s_{t'}, a_{t'}) + \sum_{a_{t+k}} \pi_1(a_{t+k}|\tau_{t+k}^i)Q^{\pi_0 \to \pi_1}(a_{t+k}|\tau_{t+k}^i) \right] \tag{3}$$

Under this setting, the states reached by $\pi_1$ may be reached at very low probabilities under $\pi_0$. The variant learned-belief OBL (Hu et al., 2021, LB-OBL) addresses this by using an approximate belief $\hat{B}_{\pi_0}$ that takes $\tau_i$ as input and samples a trajectory from an approximation of $P(\tau|\tau^i, \pi_0)$. $Q$-learning is then performed with an altered target value. Particularly, a new $\tau'$ is resampled from $\hat{B}_{\pi_0}(\tau_t^i)$. Next, a transition to $\tau_{t+1}^{i'}$ is simulated with other agents playing policy $\pi_1$. The bootstrapped value is then $\max_a Q(a|\tau_{t+2}^i)$. Hence, LB-OBL only involves fictitious transitions. The action $a_t^i \sim \pi_1$ is applied to both the actual environment and a sampled fictitious state. The learning target then becomes the sum of fictitious rewards $r_t', r_{t+1}'$ and the fictitious bootstrapped value $\max_a Q(a|\tau_{t+2}^i)$. We use exact belief models in this work to remove the influence of belief learning.

## 3.2 DIFFERENTIABLE INTER-AGENT LEARNING

Foerster et al. (2016) propose DIAL to learn how to communicate. It allows agents to give each other feedback about their communicative actions, by opening up communication channels for gradients to flow through from one agent to another. Such richer feedback improves sample complexity with more efficient discovery of protocols. During training, there are direct connections between one agent's network output and another agent's input through communicative actions. Agents can then send real-valued messages and are only restricted to discrete messages during execution. These real-valued messages are generated from the networks, allowing end-to-end backpropagation across agents.

The proposed network is called C-net, an extension of deep $Q$-networks (Silver et al., 2016, DQN). It has two set of outputs, namely the $Q$-values $Q(\cdot)$ of the environment actions $A_{env}$ and the real-valued messages $m_t^a$. The former performs actions selection while the latter is sent to another agent after the *discretize/regularize unit* $DRU(m_t^a)$ processes it. The DRU regularizes messages during learning, $DRU(m_t^a) = Logistic(N(m_t^a, \sigma))$, and discretizes them during execution, $DRU(m_t^a) = \mathbb{1}\{m_t^a > 0\}$. $\sigma$ is the standard deviation of the noise added to the cheap talk channel. The gradient term for $m$ is backpropagated to the sender based on the message recipient's error. Hence, the sender can adjust $m$ to minimize the downstream loss. Note that we use the OBL loss here instead of DQN loss.

## 4 PROBLEM FORMULATION

We consider each agent's action space $A$ to be decomposable into two subspaces, namely, the environment action space $A_{env}$ and communication action space $A_{comm}$. The former are actions that only affect the environment state. The latter are actions that have no immediate effect on the environment state or the reward received, but can alter other agents' observations. Exemplifying with PBMaze, $A_{env}$ for the sender are actions that move its locations like *Up* and *Down*, while $A_{comm}$ are actions that send messages to the receiver like *Hint Up* and *Hint Down*.

In this setting, agents do not inherently know where to best communicate and only a subset of states allows communication. We refer to this subset as the communicative state space $S_{comm} \in S$:

**Definition 4.1.** The communicative state space $S_{comm}$ of an environment is a subspace of its state space $S$. Assume the environment is in a state $s_c \in S_{comm}$, at least one agent in $s_c$ can modify another agent's observation by taking an action $a \in A_{comm}$.

Similarly, the communicative joint observation space $\mathbb{O}_{comm}$ is defined as follows:

**Definition 4.2.** The communicative joint observation space $\mathbb{O}_{comm}$ of an environment is a subspace of its joint observation space $\mathbb{O}$, defined as $\mathbb{O}_{comm} = \{\Omega(s_c)|s_c \in S_{comm}\}$. Assume the environment is in a state $s_c \in S_{comm}$ with a joint observation $o_c \in \mathbb{O}_{comm}$, at least one agent, agent $i \in N$, can modify another agent $j \in N, j \neq i$'s observation, $o_j^{t+1}$ by taking an action $a \in A_{comm}$.

With these definitions, the cheap talk discovery and utilization problem can be formalized as follows:

**Definition 4.3.** For a given $s \in S$, let $MI(s) = \sum_{i \sim N} \sum_{j \sim N, j \neq i} I(A^i, O^j)$ be a function that computes the pairwise mutual information (PMI) of an agent's actions and another agent's observations. The inner term is defined in equation 5. Let $\nu(\pi) = \mathbb{E}_\pi\left[\sum_{i=t}^{\infty} \gamma^i MI(s_t) \mid \tau_t\right]$, be the discounted sum of PMI of a policy $\pi \in \Pi$, where $\Pi$ is the set of all possible policies. Cheap talk discovery is the

problem of learning a policy $\pi_{discover}$ in which agents take actions in $A_{env}$ that maximizes $\nu(\pi)$. An optimal $\pi^*_{discover}$ satisfies the condition:

$$\nu(\pi^*_{discover}) \geq \nu(\pi) \qquad\qquad \forall \pi \in \Pi$$

**Definition 4.4.** Given a state $s_t$ is in $S_{comm}$ with a corresponding AOH $\tau_t$ at time $t$, cheap talk utilization is the problem of learning a policy $\pi_{util}$ in which agents take communicative actions in $A_{comm}$ to share information to improve task performance. For a policy $\pi \in \Pi$, $R^\pi_t = \mathbb{E}_\pi \left[ \sum_{i=t}^\infty \gamma^i r_{t+i} \mid \tau_t \right]$ is its expected return starting from $s_t$. An optimal $\pi^*_{util}$ can then be defined as:

$$R^{\pi^*_{util}}_t \geq R^\pi_t \qquad\qquad \forall_{\pi \neq \pi^*_{util}} \pi \in \Pi \text{ and } \forall \tau_t \text{ s.t. } s_t \in S_{comm}$$

We reiterate that this is a challenging joint exploration problem, especially when communication is necessary for task completion. Because agents need to stumble upon a state $s_c \in S_{comm}$, which is often a much smaller subspace than $S$, and there are no incentivizing signals to reach these states as communicative actions $A_{comm}$ do not lead to immediate rewards.

Thus, we hypothesize that this problem decomposition, i.e., first learn to discover the $S_{comm}$ before learning how to use the communicative channels, could ease this joint exploration problem's severity.

## 5 METHODOLOGY

### 5.1 CHEAP TALK DISCOVERY

Our proposed method has two components, namely, mutual information (MI) maximization and OBL (Hu et al., 2021). The former induces agents to discover cheap talk channels based on MI. The latter is our base learning algorithm with sound theoretical properties that are beneficial to the following CTU step.

#### 5.1.1 MUTUAL INFORMATION MAXIMIZATION

Agents do not have immediate incentives to discover cheap talk channels. To do so, an agent has to know where to send messages to other agents successfully (i.e., channels that can influence another agent's observations). Hence, we propose novel reward and loss functions based on MI. The proposed reward function encourages greater mutual information (MI) between the sender's (denoted as agent 1) actions $A^1$ and the receiver's (denoted as agent 2) observations $O^2$. It is expressed as:

$$R'(s_t, a_t) = R(s_t, a_t) + \beta I(A^1, O^2). \tag{4}$$

where $\beta$ is a hyperparameter. The first term is the task reward. The second term is the MI reward:

$$I(A^1, O^2) = \mathop{\mathbb{E}}_{\substack{a^1 \sim A^1 \\ o^2 \sim O^2}} \left[ \log(p(a^1|o^2) - \log(p(a^1))) \right] \tag{5}$$

We assume access to the environment simulator to estimate $p(a^1|o^2)$. The proposed reward function can then be substituted in the update equation 3. To estimate $p(a^1|o^2)$, the term can be expanded to:

$$
\begin{aligned}
I(A^1, O^2) &= H(A^1) + H(O^2) - H(A^1, O^2) \\
&= \sum_{a^1 \sim A^1} \left[ -p(a^1) \log(p(a^1)) \right] + \sum_{o^2 \sim O^2} \left[ -p(o^2) \log(p(o^2)) \right] \\
&\quad - \sum_{\substack{(a^1, o^2) \\ \sim (A^1, O^2)}} \left[ -p(a^1, o^2) \log(p(a^1, o^2)) \right]
\end{aligned}
\tag{6}
$$

where $p(o^2) = \sum_{a^1 \sim A^1} [p(a^1, o^2)]$ and $p(a^1, o^2) = p(o^2|a^1)p(a^1)$. Note that this generalizes to any pair of agents in an environment irrespective of the receiver. Hence, we can apply this method to all possible receivers by summing all $I(A^1, O^i), i \in N, i \neq 1$ to discover the cheap talk channels.

The MI reward term alone does not maximize $I(A^1, O^2)$. Specifically, for a given policy, since the term is computed based on taking all the actions from a state, the MI reward for staying within

$S_{comm}$ and for taking an action in $A_{comm}$ are equal. Hence, a policy learned only with MI reward is incentivized to reach $S_{comm}$ without necessarily using communicative actions. Ideally, $I(A^1, O^2)$ is maximized when a policy favors the communicative actions equally when in $S_{comm}$.

To learn a policy that does maximize $I(A^1, O^2)$, we propose an MI loss function, using the parameterized policy $\pi_\theta$. The additional term directly maximizes $I(A^1, O^2)$, which is maximized when the policy takes actions in $A_{comm}$ more. For each iteration $i$, the loss function is expressed as:

$$L_i(\theta_i) = L_i^{OBL}(\theta_i) - \kappa I(A^1, O^2; \pi_\theta)_i \tag{7}$$

where $L_i^{OBL}(\theta_i)$ (see Appendix B), $I(A^1, O^2; \pi_\theta)_i$ and $\kappa$ are OBL loss, MI loss at iteration $i$, and a hyperparameter respectively. We add a minus sign so that minimizing the loss maximizes the MI term. The combination of the MI reward and loss should allow the discovery of cheap talk channels with greater MI. We denote our discovery method as *cheap talk discovery learning (CTDL)*

### 5.1.2 DISCOVERING CHANNELS WITHOUT CONVENTIONS FORMATION

During discovery learning, agents should simply learn where the channels are without forming any protocols (i.e., conventions) on how to use them. This is because the protocols formed would affect the rewards received. To do so, we employ OBL as our base algorithm.

Particularly, it has appealing theoretical properties which are beneficial for this setting, as it theoretically guarantees to converge to a policy without any conventions (Hu et al., 2021). This means OBL with CTDL would learn a policy that discovers the channels while not having a preference over any communicative actions, which would allow more flexible adaptation when properties of channels alter. Hence, though not explored here, better performance in ZSC can be expected.

For our example environment, the agent should prefer the actions *Hint Up* and *Hint Down* equally after cheap talk discovery. If the agents are in a slightly different environment with the phone booths sending negated versions of the messages an agent sends, a policy without conventions should adapt quicker given that it has no preference over communicative actions.

### 5.2 CHEAP TALK UTILIZATION

With channels discovered, agents are ready to learn how to use them through protocol formation (i.e., form a consensus of how messages are interpreted). Many off-the-shelf algorithms can be used for CTU. Here, we use DIAL (Foerster et al., 2016), which has been shown to learn protocols efficiently. Note that OBL will be replaced by independent Q-learning (Tan, 1993, IQL) updates to allow protocol formation in CTU.

Given that the original DIAL requires feedforwarding through an episode's full trajectory to maintain the gradient chain, modifications to DIAL are needed to work in this setting for two reasons:

First, the base learning algorithm is an experience-replay-based method that learns from transition batches, not trajectories. Second, unlike in the original DIAL which messages are sent in every step $t$, the messages can only be sent successfully when the environment state is in $S_{comm}$. Hence, direct gradient connections only happen occasionally in an episode.

To make DIAL compatible with this setting, we use a separate replay buffer to keeps track of the transitions and only consider a trajectory when messages are sent successfully. The buffer stores full episodes with flags indicating which transitions to allow direct gradient connections. We denote our discovery method with DIAL as ***cheap talk discovery and utilization learning*** (CTDUL). The transition from CTD to CTU is a task-dependent hyperparameter and is determined empirically. Appendix C and D shows how gradients flow in CTDUL and the corresponding pseudocode.

## 6 EXPERIMENTAL SETUP

As part of our contribution, we designed a configurable environment with discrete state and action space - *the phone booth maze* to evaluate agents in the CTD/CTU setting, as described in Section 1.

To enable extensive investigations, the environment allows phone booths of configurable properties including usage cost and noise level, and decoy booths that are not functional. The agents' observation

contains view information of the room an agent is in and role-specific information (e.g. the sender and receiver have goal information and communication token respectively). Agents have the environment actions of *UP*, *DOWN*, *LEFT*, *RIGHT*, and *NO-OP*. There are two communicative actions, *HINT-UP* and *HINT-DOWN* which only has an effect when agents are in the functional phone booths, updating the receiver's communication token to a fixed value. Rewards of 1.0 and -0.5 are given for choosing a correct exit and an incorrect exit respectively with no reward given otherwise. By changing the phone booths' properties, we can vary their capacities, measured by the MI reward.The left of Figure 2 shows the MI of each location in the sender's room in two different environment configurations, in which one has only one functional booth and the other has three booths with one noisy booth. These are computed based on equation 5 on all possible combinations of the sender and receiver locations.

We use two configurations of the environment for our experiments, which we name Single Phone Booth Maze (SPBMaze) and Multiple Phone Booth Maze (MPBMaze). Please see Appendix E for environment visualizations and design features, and Appendix H for environment parameters used.

Architecturally, we use the R2D2 architecture (Kapturowski et al., 2018) with a separate communication head like in C-Net (Foerster et al., 2016). For baselines, state-of-the-art methods are used including IQL, OBL (Hu et al., 2021), QMIX (Rashid et al., 2018), MAA2C (Papoudakis et al., 2021) and IQL with social influence reward (Jaques et al., 2019, SI).

We trained all methods for 12000 episodes (80000 episodes for CTU) and evaluated on test episodes every 20 episodes by taking the corresponding greedy policy. Each algorithm reports averaged results from 4 random seeds with standard error. We performed a hyperparameter sweep over common hyperparameters, fixing them across all methods, and specific sweeps for method-specific parameters. Please see Appendix F and G for training and hyperparameter details.

# 7 RESULTS

## 7.1 DISCOVERING CHEAP TALK CHANNELS

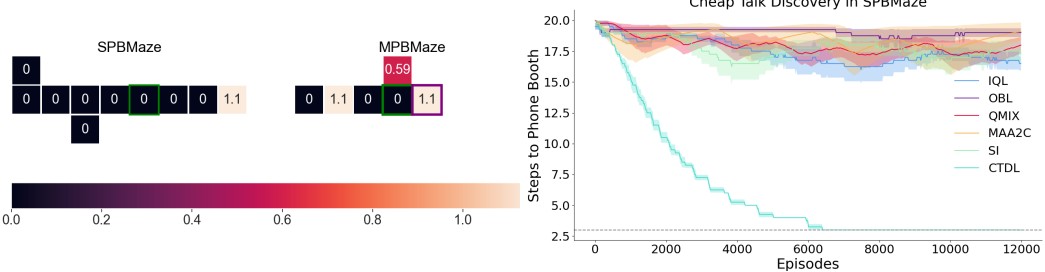

Figure 2: Left: MI reward for each location in the sender's room of SPBMaze and MPBaze to show phone booths of different properties. Values are computed by iterating over all possible receiver's position for each sender's position. Senders start at positions with green borders. The position with purple border is the costly phone booth which incurs a cost to use. The top phone booth in MPBaze has a noise factor of 0.5, i.e., lower MI. Right: CTD performance for CTDL and baselines. The horizontal grey line indicates the optimal number of step to reach the booth. For the ease of plotting, when an agent cannot discover the booth, we set the step as the episode length.

To quantify the performance in CTD, we use the number of steps it takes for the sender to first reach the functional phone booth in SPBMaze. The fewer the number of steps used, the better an algorithm is at CTD. See Table 4 for detailed configuration of SPBMaze.

Figure 2 (Right) shows the performances of our proposed method CTDL and the baselines. CTDL, using MI reward and loss, discovers the functional phone booth quickly in optimal number of steps (as shown by the horizontal grey line). Meanwhile, without the proper incentives, all the baseline methods cannot discover the functional phone booth within the episode limit.

Even though SI uses a reward term based on MI like ours (i.e., the influence an agent's message has on another agent's action (Jaques et al., 2019)), it becomes insufficient to discover cheap talk

channels in this setting as the messages sent only has an effect in a small subset of the state space. Note that without discovering the channels, agents will not be able to learn how to communicate. Hence, we expect the baseline methods cannot leverage communication channels to solve tasks.

## 7.2 LEARNING POLICIES WITH MAXIMIZED MUTUAL INFORMATION

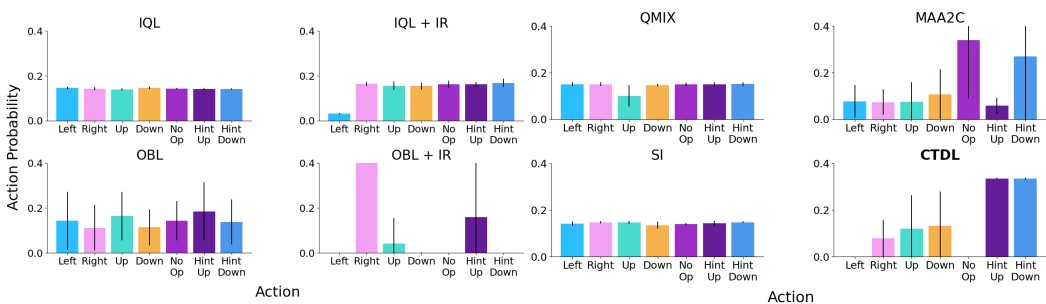

Figure 3: Algorithms' sender policy when both agents are at the functional phone booth in SPBMaze. CTDL learns a policy with the most MI by preferring communicative actions equally.

To assess whether an algorithm can learn policies that maximize MI (i.e., $I(A^1, O^2)$) in this setting, we look at the sender's policy when both agents are at the functional phone booth. To reiterate, such policy uniformly prefers $A_{comm}$ over $A_{env}$ when using a communication channel without forming any protocol. Figure 3 shows different algorithms' sender policy when in the functional phone booth.

We include two variants that receive a scalar intermediate reward (IR) when both agents are at functional phone booths. As discussed in section 5.1.1, these reward shaping methods do not maximize MI, although cheap talk channels are discovered. Instead, they learn to prefer actions that keep them in $S_{comm}$ which includes certain environment actions.

Contrarily, our proposed MI loss used in CTDL leads to polices with the most MI by preferring $A_{comm}$ (i.e., *Hint-Up* and *Hint-Down*) uniformly over $A_{env}$. This explains why CTDL has the best CTD performance as it directly maximizes PMI in Definition 4.3. Note that this uniformity over $A_{comm}$ also means no conventions are formed during discovery learning as they are preferred equally under different goals (i.e. different correct exits in SPBMaze), demonstrating the effect of OBL.

## 7.3 UTILIZING CHEAP TALK CHANNELS

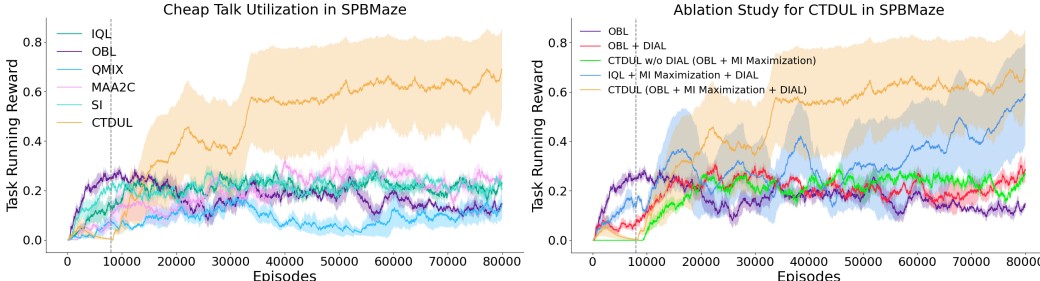

Figure 4: Left: CTU performance for CTDUL and baselines on SPBMaze. The vertical grey line indicates the start of CTU learning for CTDUL. Right: Ablation experiment on SPBMaze for CTDUL. The curves are smoothed by a factor of 0.99 with standard errors plotted as shaded areas.

Figure 4 (left) shows the overall task performance of the proposed CTDUL and baselines in solving the SPBMaze. Being unable to communicate the goal information successfully, the receiver can only guess by escaping randomly (i.e., achieving an expected reward of $0.25 = 1.0 \times 0.5 + (-0.5) \times 0.5$). Hence, all baselines converge to the suboptimal solution of random guessing.

From Figure 2 (right) and Figure 4 (left), we can infer that the senders in the baselines do not reach the functional phone booth to communicate so no protocols can be formed. Centralized training baselines

(e.g., QMIX, MAA2C), which are designed to assign credits better since they have centralized components to learn better value functions using full state information, also converge to suboptimal solutions. Their poor performances illustrate the difficulty of this joint exploration problem.

In contrast, our proposed method CTDUL obtains significantly higher rewards than random guessing by employing DIAL to learn how to use the discovered channel. The grey vertical line is when DIAL begins after discovery learning. Qualitatively, we observe how our method solves the problem in which the receiver waits at the functional booth until receiving a message from the sender. Then, it goes to the correct exit according to the protocol learned. The results show how our proposed problem decomposition and MI maximization components make the difficult problem easier to solve.

Figure 4 (right) shows task performance on SPBMaze for our ablation study with CTDUL. Specifically, we look at the task performance when we individually remove components used, namely MI maximization and DIAL. As we can see, nontrivial solutions can only be learned if both of them are used. Removing one of them leads to the suboptimal solution of random guessing. Furthermore, we observe significant performance dip when OBL is replaced with IQL, empirically supporting the importance of not forming any conventions during CTD as pointed out in section 5.1.2.

## 7.4 MUTUAL INFORMATION AS A MEASURE OF CHANNEL CAPACITY

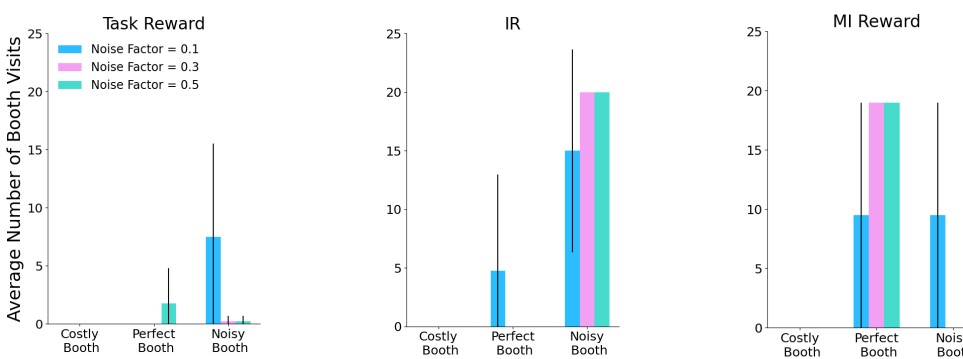

Figure 5: Bar plot of average number of booth visits of each phone booth type in MPBMaze with varying noise factors for the noisy booth. The error bars are standard errors for the average number of booth visits for each booth type.

We hypothesize that our proposed MI reward can be a pseudomeasure of channel capacity, a way to discover where best to communicate when an environment has multiple channels. To verify, we run baselines and our proposed method for CTD on MPBMaze which has three channels, namely, the Perfect Booth, the Costly Booth, and the Noisy Booth. Note that a booth with noise has a probability of dropping a message. See Table 4 in the Appendix for the detailed configuration of MPBMaze.

Figures 5 compare which booth the sender chooses to visit when different reward functions are used. The optimal behavior is to visit the Perfect Booth the most as it has no noise and usage cost. All algorithms learn to avoid the Costly Booth given its direct effect on the reward. Comparing methods that use IR and MI rewards, the IR reward cannot distinguish between the Perfect Booth and the Noisy Booth and consistently visit the Noisy Booth as visits to both booths are rewarded equally and the Noisy Booth is closer to the sender.

On the contrary, the MI reward method is able to visit the Perfect Booth consistently as the MI reward has an interpretation and measurement of noise and channel capacity (i.e., A channel with greater noise would necessarily mean a lower expected MI reward). As the noise factor decreases, the two booths become less distinguishable, which we observe a decreased in number of booth visits to the perfect booth for the agent with MI reward when the noise factor is 0.1.

## 8 CONCLUSION

In this work, we take away the common and arguably unrealistic assumptions in MARL that communication channels are known and persistently accessible to the agents. Under this setting, agents have

to first discover where to best communicate before learning how to use the discovered channels. We first provide a novel problem formulation for this setting denoted as the *cheap talk discovery and utilization problem*. Then, based on this problem decomposition, we propose a method based on MI maximization with OBL and DIAL to tackle this problem end-to-end. Experimentally, by evaluating our framework against state-of-the-art baselines in our proposed environment suite, we show that our framework can effectively discover communication channels and subsequently use them to solve tasks, while all the baselines fail to do so. We also empirically attribute the framework's success to its key capabilities in learning policies with maximized MI and without convention formation during CTD. Finally, we demonstrate how our MI metric can serve as a pseudomeasure for channel capacity, to be used to learn where best to communicate. We hope this work can inspire investigation in more realistic settings for communication like the one introduced here.

## 9 ACKNOWLEDGMENTS

We thank Tarun Gupta for providing code implementations of RL environments for our reference during the ideation phase of the project. We thank Michael Noukhovitch for helpful feedback.

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

## A    PROBLEM SIGNIFICANCE AND USE CASES

The full problem setting of solving Dec-POMDPs is NEXP-complete. As such, it is not reasonable to expect a method that can solve large arbitrary Dec-POMDPs. To get around this, the "emergent communication" literature usually focuses on a setting where there is a fixed communication channel that is known a-priori.

We strictly generalise this setting by instead proposing a method that can first discover channels within the environment and next utilise them. Clearly, this cannot cover all problem Dec-POMDPs due to the constraints mentioned above but includes a broader set of potentially practically relevant applications.

In terms of use cases, the problem applies whenever real-world communication through a medium among agents is considered. Because more often than not, there will be some spatial constraints on communication in realistic settings. Any kind of communication channel used in the real-world (e.g., a radio link, cell phone signal), does not have perfect signal strength, and often transmission errors could happen (e.g., noise). The signal strength varies a lot depending on your environment and current surroundings (e.g, indoor vs outdoor, underwater, environment interference in certain locations like behind a mountain). Hence, for communication to happen efficiently and successfully, being able to learn where to communicate and learn where best to communicate is crucial which our problem setting addresses.

Even in communication without an explicit medium, say if a robot wants to signal to another one with lights but they need a line of sight, communication becomes spatially constrained. We believe these examples illustrate how important this problem setting is when considering communication learning in more realistic environments.

Furthermore, the setting also sheds light to a more complete picture of capacity-constrained communication. As illustrated in our experiments with noisy channels, our mutual information reward provides a pseudomeasure of capacity. The investigation provides a path forward regarding capacity-constrained communication, by establishing two levels of capacity constraints - i.e., capacity constrained by the environment/physical space, size of $S_{comm}$, and the capacity for individual channels. Our method provides the first attempt to tackle both types of capacity constraints - **_learning where to communicate_** and **_learning where best to communicate_**.

## B    OBL LOSS

As in Hu et al. (2021), for a given trajectory $\tau$ sampled from replay buffer, $L^{OBL}$ is expressed as TD-error:

$$L^{OBL}(\theta|\tau) = \frac{1}{2} \sum_{t=1}^{T} \left[ r'_t + r'_{t+1} + \max_a Q_{\hat{\theta}}(a|\tau'_{t+2}) - Q_{\theta}(a_t|\tau_t) \right]^2 \tag{8}$$

where $Q_{\hat{\theta}}$ is the target network. As $\tau'_t$ contains fictitious transitions, we can not pass the whole sequence to RNN like in normal Q-learning. Thus, we precompute the target $G'_t = r'_t + r'_{t+1} + \max_a Q_{\hat{\theta}}(a|\tau'_{t+2})$ during rollouts and store the sequence of targets along with the real trajectory $\tau$ into the replay buffer.

## C    GRADIENT CHAIN OF THE PROPOSED ARCHITECTURE

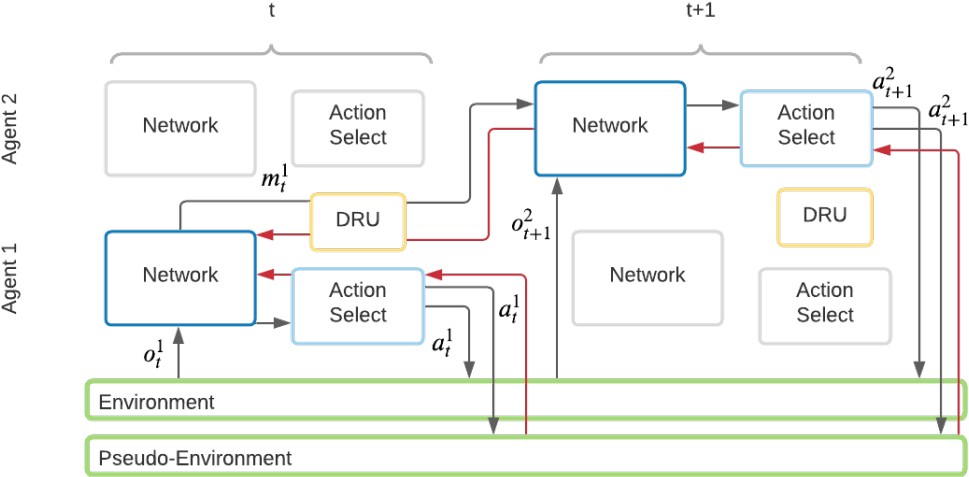

Figure 6: How gradients flow through our proposed architecture, with reference to Foerster et al. (2016)

## D    PSEUDOCODE OF THE PROPOSED ARCHITECTURE

Algorithm 1 outlines the pseudocode of the proposed architecture to tackle the cheap talk discovery and utilization problem.

---

**Algorithm 1** Pseudocode for our proposed method

---

**for** each agent **do**
    Initialize replay memory for OBL $D_{OBL}$ with capacity $N$
    Initialize replay memory for DIAL $D_{DIAL}$ with capacity $N$
    Initialize action value function Q with random weights $\theta$, using architecture C-Net based on R2D2
    Initialize target action value function $\hat{Q}$ with random weights $\theta^- = \theta$ , using architecture C-Net based on R2D2
    Initialize Num_Discovery_Episode
    Initialize Discovery_Stage as True
**end for**
**for** e = 1, Max_Episode **do**
    Initialize $s$
    **repeat**
        **for** each agent **do**
            Select $a$ from $o$ based on policy derived from $Q$, e.g. $\epsilon$-greedy policy or stochastic OBL policy
            Take action $a$ to observe reward $r$ and next state $o'$
            Take action $a$ in Pseudo-Environment to perform OBL sampling with mutual information computation
            Store the transition into $D_{OBL}$
            **if** High mutual information is observed **then**
                Store trajectory into $D_{DIAL}$
            **end if**
            Sample random minimatch of transitions from $D_{OBL}$
            Sample random minimatch of transitions from $D_{DIAL}$
            **if** Discovery_Stage **then**
                Perform a gradient descent step on $L_i^{OBL}(\theta_i) - \kappa I(A^1, O^2; \pi_\theta)_i$ with respect to the network weights $\theta$
            **else**
                Perform a gradient descent step on $(y_i - Q(o, a; \theta_{i-1}))^2$ for DIAL with respect to the network weights $\theta$
                Perform a gradient descent step on $L_i^{IQL}(\theta_i)$ with respect to the network weights $\theta$
            **end if**
            Every C steps, set $\hat{Q} = Q$
        **end for**
    **until** $s = s_{terminal}$
    **if** e $\geq$ Num_Discovery_Episode **then**
        Set Discovery_Stage as False
    **end if**
**end for**

---

## E   Environment Design

Figure 7 visualizes two instances of the environment used in our experiments. Red cables indicate connectivity with dashed ones indicating the existence of noise. The yellow booth here indicates that it is a costlier booth to use.

For observations, each consists of a tensor of 3 channels plus some role-specific information. The channels are the wall channel, phone booth channel, and the agent channel, which are essentially binary grid encoding of each position for the existence of wall, phone booth, and the agent respectively. For role-specific information, the sender has an additional 2-bit encoding vector as goal information. Specifically, if the receiver should go up, the sender would get a vector of $[1 \quad 0]$. If the receiver should go down, the sender would get $[0 \quad 1]$ instead. For the receiver, if the sender performs a *HINT-UP* with both of them at the functional booths, it would be -1. On the other hand, if the sender performs a *HINT-DOWN*, it would be 1. Many aspects of this environment are configurable to allow extensive investigation and exploration of different algorithms, some key adjustable features are highlighted in table E.

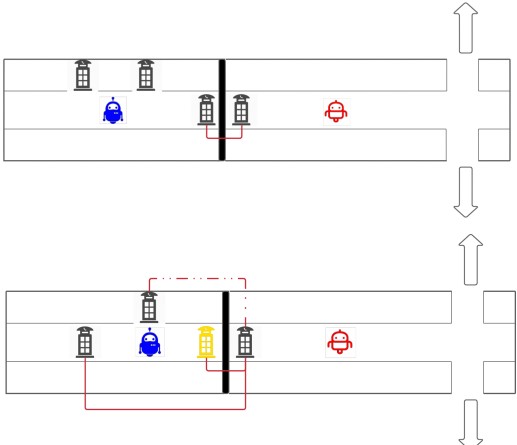

Figure 7: Two instances of the PBMaze with different configurations used in the experiments

## F    NETWORK ARCHITECTURE AND TRAINING

Instead of using convolutional layers as in Kapturowski et al. (2018), we use fully-connected neural networks with a recurrent component as our neural network model. Hence, we flatten the observation from the environment into a vector by first fattening the 3-channel tensor and then concatenate it with the role-based information (i.e., goal encoding or communication token). Precisely, the input is first processed by an LSTM layer to handle partial observability. Then, it is followed by a two-layer and two-headed fully-connected neural network of hidden size 128. The two heads are used to compute the action-value function and the advantage function respectively. All neural network components are implemented using the neural network library PyTorch (Paszke et al., 2019), with the weights initialized using Xavier initialization. In terms of training, we use the Adam optimizer to train our models (Kingma & Ba, 2014). Training was done in an internal cluster with a mix of GTX 1080 and RTX 2080 GPUs.

## G    HYPERPARAMETERS SETTING

This section covers details in setting hyperparameters for various methods used in this work. They are covered in two separate sections depending on whether a parameter is common to all used methods or not.

### G.1    COMMON PARAMETERS

To determine the best parameters that are common to all the methods used, we performed a hypa-rameter sweep over some key common parameters. Specifically, the search was performed on the learning rate and target network update frequency in the lists of $[0.01, 0.001, 0.0001, 0.00001]$ and $[50, 100, 200, 500]$. The results are averaged over 3 random seeds, each trained for 25000 episodes. The best set of parameters for learning rate and target network update frequency are 0.0001 and 100, respectively. Other common parameters are set to values in table G.1.

### G.2    METHOD-SPECIFIC PARAMETERS

Method-specific parameters are set to values in table 3.

## H    ENVIRONMENT PARAMETERS

Parameters of the environments used in the experiments are summarized in Table 4.

| Configurable parameter | Description |
|---|---|
| lengths | the length of each of the agent's rooms. The longer it is, the harder the exploration becomes. |
| starting points | the starting location of each of the agents. |
| correct reward | The reward that is given when the receiver exits from the correct door. |
| wrong reward | The reward that is given when the receiver exits from the wrong door. |
| use intermediate reward | If set as true, it gives an intermediate reward for if both agents are at the functional booths, used for debugging purposes. |
| episode limit | Episode length, meaning the episode terminates if it reaches this number of steps. |
| booth types | Among functional phone booths, they are configurable in two major ways, namely, cost and noise. The former refers to the cost of using the booth and the latter is modeled as the probability of a phone booth dropping a message. |
| booth locations | Locations of functional booths. |
| number of decoy booths | The number of decoy booths in the sender's room. Decoy booths refer to booths that are not functional and not connected to the booth in the receiver's room. They appear as booths in the agent's observation (i.e., the booth channel). |
| booth reinitialization | If set true, the decoy phone booths' locations are reinitialized whenever the environment is reset for the next episode, making the problem more difficult. |
| use mutual information reward | If set true, the environment would compute the mutual information reward, used in the proposed method. |
| use mutual information loss | If set true, the environment would return the necessary information (i.e., tensor masks for each term in equation 6 to compute the mutual information loss in a batch-based manner. |

Table 1: Configurable parameters for the PBMaze

| Hyperparameter | Value |
|---|---|
| Discount Factor $\gamma$ | 0.99 |
| Batch Size | 32 |
| Replay Buffer Size | 10000 |
| Temperature | 1.0 |
| Non-Linearity | ReLU |

Table 2: Common parameters used across algorithms

## I  LIMITATIONS

While our proposed framework provides a promising approach to learn where to communicate (i.e., CTD/CTU) by first discovering cheap talk channels and subsequently learning how to use them, it is important to be aware of the assumptions and limitations of the framework for future work.

To begin with, our approach assumes access to an environment simulator and a (perfect) belief to perform MI computation and OBL respectively. While there are recent approaches in learning environment models (Wang et al., 2022) and belief models (Hu & Foerster, 2019), how well these learned components work with each other remains unexplored. Advances in these areas would improve the overall robustness of our proposed framework.

| | IQL | IQL + IR | OBL | CTDL/CTDU | QMIX | MAA2C | SI |
|---|---|---|---|---|---|---|---|
| Starting $\epsilon$ | 1.0 | 1.0 | N/A | N/A | 1.0 | 1.0 | 1.0 |
| $\epsilon$ Decay Step | 0.00001 | 0.00001 | N/A | N/A | 0.00001 | 0.00001 | 0.00001 |
| Minimum $\epsilon$ | 0.1 | 0.1 | N/A | N/A | 0.1 | 0.1 | 0.1 |
| Initial Exploration Step | 1000 | 1000 | 1000 | 1000 | 1000 | 1000 | 1000 |
| IR | N/A | 1.0 | N/A | N/A | N/A | N/A | N/A |
| $\alpha$ Entropy Factor | N/A | N/A | N/A | 0.0 | N/A | N/A | N/A |
| $\beta$ Mutual Information Reward Factor | N/A | N/A | N/A | 2.0 | N/A | N/A | N/A |
| $\kappa$ Mutual Information Loss Factor | N/A | N/A | N/A | 1.0 | N/A | N/A | N/A |
| learning rate | N/A | N/A | N/A | N/A | 0.0005 | 0.0001 | N/A |
| critic learning rate | N/A | N/A | N/A | N/A | N/A | 0.0001 | N/A |
| critic hidden size | N/A | N/A | N/A | N/A | N/A | 32 | N/A |
| hypernetwork hidden size | N/A | N/A | N/A | N/A | 64 | N/A | N/A |
| number of hyper layers | N/A | N/A | N/A | N/A | 1 | N/A | N/A |
| social influence reward coefficient | N/A | N/A | N/A | N/A | N/A | N/A | 5.0 |
| temperature | 1.0 | 1.0 | 0.1 | 0.1 | 1.0 | 1.0 | 1.0 |

Table 3: Method-specific parameters

Another crucial assumption made is the direct and immediate causal relationship between a communicative action and a receiver's observation which might be less reliable in the real-world setting. For instance, there will be delay and transmission time when sending messages. In more visually-rich and open-ended settings, there could also be many confounding changes in the receiver's observation when communication actions are taken. In these cases, further advances are needed to correctly assign credits to communicative actions when computing MI.

Last but not least, the transition from CTD to CTU is determined empirically from experiments. A more principled approach to automate this decision would make method more robust. For instance, based on Definition 4.3, a method could be developed to indicate whether an agent has been able to discover a channel consistently (i.e., a discovery policy with maximized PMI). Once a threshold is reached, it can move on to the utilization stage.

| | SPBMaze | MPBMaze |
|---|---|---|
| lengths (sender's room, receiver's room) | (8, 4) | (5, 3) |
| starting points (sender's coordinates, receiver's coordinates) | ((4, 1), (2, 1)) | ((3, 1), (1, 1)) |
| correct reward | 1.0 | 1.0 |
| wrong reward | -0.5 | -0.5 |
| episode limit | 20 | 20 |
| booth types | 1 functional booth, 0 cost to use and 0 noise factor | functional phone booths. The first one has a cost of 0.4 and 0 noise factor. The second one has 0 cost and 0 noise factor. The third one has 0 cost and $x \in [0.1, 0.3, 0.5]$ as noise factor across 3 sets of experiments |
| number of decoy booths | 2 | 0 |
| booth reinitialization | False | False |

Table 4: Table for environment configurations of each environment used in the experiments

## J  BROADER IMPACT

With the rapid progress in MARL and multi-agent learning systems in general, we confidently expect that more multi-agent learning systems will be deployed in the real world in which communication will play an integral part to their successes. Our formulation and method for learning where to communicate would lead to more robust communication and subsequently more coordinated behavior among agents by equipping them with the ability to discover the best and most reliable places to communicate. This is essential in such real-world applications when the reliability of communication often varies by locations and situations.

By increasing the applicability of multi-agent learning systems, our work may exacerbate some risks of deploying machine learning systems like increasing unemployment (e.g., replacing warehouse workers with a fully autonomous warehouse) and advancing automated weaponry. Particularly to our method, by equipping systems with the ability to discover where to communicate, it could encourage adversarial attacks to deployed machine learning systems through communication which would lead to unintended and potentially harmful actions and disrupt learned communication protocols.

## K  FUTURE WORK

For future work, we are interested in testing our framework in more complex settings and address some limitations discussed in Appendix I. For instance, the ability to discover communication channels can be formulated as skills and organized in a hierarchical manner to solve harder tasks. It would also be insightful to benchmark different algorithms for cheap talk utilization in this setting.

