# OpenReview forum: "Cheap Talk Discovery and Utilization in Multi-Agent Reinforcement Learning"
_ICLR.cc/2023/Conference — ICLR 2023 poster_

### Official Review · Reviewer_4J9r · 2022-10-20

**Confidence:** 3
**Correctness:** 4
**Technical Novelty And Significance:** 3
**Empirical Novelty And Significance:** 3
**Recommendation:** 8

**Clarity, Quality, Novelty And Reproducibility:**

# Quality
The combination of OBL and DIAL is technically novel. It is interesting that OBL is used specifically due to its inability to form a protocol (as to avoid forming an inappropriate protocol in the early stages.)  It would have been very interesting to extend to more agents, but only two agents are considered here. The two examples include cost and noise parameters to provide more illustration of the method.

# Clarity
The paper is generally written clearly.

# Originality
The problem of learning to communicate when very deliberate actions are needed to enable communication does not seem to be studied in the literature. There exist environments in the literature where communication is bounded by spatial constraints, but where it is relatively easy for agents to accidentally enter communication-enabling states. But it seems rather more realistic to assume that communication is difficult, and this work considers that possibility.

**Strength And Weaknesses:**

+ Interesting problem
+ Novel approach of separating discovery and utilization, which seems justified
+ Thoughtful choice of combination of learning methods
- Only applied to two agents (show an example with more agents)
- States that the number of iterations in discovery stage is a "tuning parameter", but doesn't give any easy way to adjust this other than trial and error.  (Give more guidance how to set number of iterations, e.g. provide a testing procedure for whether sufficiently many iterations has been reached, or comment on such a possibility.) [after rebuttal: authors are looking into this]

**Summary Of The Paper:**

This paper demonstrates that current methods in multi-agent reinforcement learning (MARL) do not suffice for a scenario where communication between agents is possible--but only in a subset of the state space. For a case of two agents, it proposes a method of training MARL agents via a two-stage process of first learning a policy that will enable communication and then learning an optimal policy starting from a communication-enabling state. The method successfully learns to enable cooperative problem solving in a simple environment.

Note: I previously reviewed this paper as a NeuRIPS submission.

**Summary Of The Review:**

This is an interesting paper for the MARL field. There is currently a disconnect between the real-world (where communication may be limited) and MARL scenarios which make strong assumptions on the possibility of communication. This paper raises the interesting question of what can be done when communication is difficult. Furthermore, if the question of how to enable communication are further investigated, this might have implications for theories of the socio-biological origins of culture and language.

Thus, I recommend acceptance. [no change to recommendation after rebuttal]

---

> ### Author Response · Authors · 2022-11-10
> **Response to Reviewer 4J9r**
>
> We would like to thank the reviewer for the helpful comments and critiques. We fully agree with the review’s comment on the disconnect between the real-world and MARL scenarios. One of the main motivations of this work is to close this gap in the subfield of MARL communication by removing some of the unrealistic assumptions about communication channels. We hope this work could encourage more investigations in more realistic MARL settings. We also share similar views on how investigations in more realistic MARL communication settings could have important and useful implications in the study of language in general.
>
> > **Only applied to two agents (show an example with more agents)**
>
> We fully agree that having experiments with more agents will provide further insights. For future work, we plan to develop an environment with more agents. However, we emphasize that we limited our experiments to two agents to conduct a more in-depth analysis, and our proposed method can in principle generalize to any number of agents as the sender does not have to distinguish between receivers to discover communication channels.
>
> > **States that the number of iterations in discovery stage is a "tuning parameter", but doesn't give any easy way to adjust this other than trial and error. (Give more guidance how to set number of iterations, e.g. provide a testing procedure for whether sufficiently many iterations has been reached, or comment on such a possibility.)**
>
> There are potential ways to automate this. This can be framed as measuring learning progress. In this case, it would be the progress of learning where to communicate. One idea could be having an exponential moving average of the MI reward. If it is significantly greater than zero consistently enough (because nonzero MI reward can only happen when communication is successful), we can initiate the transition. We could maybe do something similar using the MI loss too. Does this seem plausible to you? We are looking into these ideas and will report back if we have something substantial before the end of the review period.

---

> > ### Comment · Reviewer_4J9r · 2022-11-16
> > **Good idea**
> >
> > > One idea could be having an exponential moving average of the MI reward. If it is significantly greater than zero consistently enough (because nonzero MI reward can only happen when communication is successful), we can initiate the transition. We could maybe do something similar using the MI loss too. Does this seem plausible to you? We are looking into these ideas and will report back if we have something substantial before the end of the review period.
> >
> > I would be interested in seeing if this works.

---

> > > ### Author Response · Authors · 2022-11-18
> > > **Follow-up on discovery-utilization transition**
> > >
> > > We have made an attempt in using the exponential moving average of the MI reward to automatically determine the transition. Please see this anonymous [link](https://postimg.cc/PCkwZ5tL) for the plot. Here we use the episodic MI reward averaged over timestep in an evaluation episode. As discussed, the threshold is automatically selected by the first MI reward that is significantly greater than zero. To update the exponential moving average, we use a multiplier of 0.98 as such:
> > >
> > > $mi\\_reward\\_ema = 0. 98 \times previous\\_mi\\_reward\\_ema + 0.02 \times mi\\_reward$
> > >
> > > The blue line (mi_reward_ema) is the exponential moving MI reward. The red line (hard_coded) is 8000th episode mark we use in the paper. The green line is the transition point determined by this exponential moving average approach (It triggers a transition at the 7475th episode mark). The purple line is the automatically determined threshold.
> > >
> > > Although being a preliminary attempt, it shows the viability of directly using the MI reward to determine the transition between discovery and utilization. The automatic approach transitions at a point similar to the one we hand-picked. Does this seem like a good approach to you? We will try to do more testing and polish the results a bit before adding this part into the camera-ready/next version of the paper.

---

### Official Review · Reviewer_Hktd · 2022-10-21

**Confidence:** 4
**Correctness:** 3
**Technical Novelty And Significance:** 4
**Empirical Novelty And Significance:** 3
**Recommendation:** 6

**Clarity, Quality, Novelty And Reproducibility:**

Clarity: Need to improve

Quality: Experiments show that their method has significant effects

Novelty: The authors have studied a novel setting

Reproducibility: Author's open source code is required

**Strength And Weaknesses:**

Strength.
* The authors propose a more realistic setting
* The authors provide a corresponding experimental environment
* Experiments show that their approach has significant advantages

Weaknesses:
* The notation used in section PROBLEM FORMULATION is cumbersome and the overall definition is redundant
* The method seems to be relatively simple, and it is surprising why it works so well, and I hope the author will open source code subsequently.
* Because in the current setting, the communication action directly affects part of the information in obs, does this mean that it is easier to find the communication status by MI? Can such an approach be extended to other scenarios?

**Summary Of The Paper:**

The authors propose a more realistic multi-agent communication setup and provide a corresponding experimental environment. The authors also propose a corresponding learning framework based on MI to solve the corresponding communication problem, and the experiments show that their approach has significant advantages.

**Summary Of The Review:**

The author has researched a novel and interesting scene, but the writing as a whole needs further improvement.

---

> ### Author Response · Authors · 2022-11-10
> **Response to Reviewer Hktd**
>
> We are grateful for your valuable feedback and helpful comments. We will address your questions and concerns below
>
> > **The notation used in section PROBLEM FORMULATION is cumbersome and the overall definition is redundant**
>
> Can you provide examples of places in the problem formulation that you find cumbersome?  The formulation is a direct extension of Dec-POMDP and only uses extra notations necessary to fully describe the problem decomposition. We also respectfully disagree with the reviewer’s view that the definitions are redundant. Given that this is a novel setting in MARL with a natural problem decomposition (i.e. cheap talk discovery and cheap talk utilization), we think it is crucial to define what would be the conditions and optimal policies in solving the problem, similar to how an optimal policy is defined when RL is first introduced [1]. This gives an unambiguous measurement tool for assessing a solution. Besides, as the setting inherently has two subproblems, not introducing new notations would make it even more difficult to understand, especially in separating environment actions and communicative actions.
>
> [1] Sutton, Richard S., and Andrew G. Barto. Reinforcement learning: An introduction. MIT press, 2018.
>
> > **The method seems to be relatively simple, and it is surprising why it works so well, and I hope the author will open source code subsequently.**
>
> Although the proposed method leverages two existing works (i.e., OBL and DIAL) with MI maximization, we respectfully disagree that the method is simple. OBL is a careful design choice given our understanding of the problem that no protocols should be formed during cheap talk discovery. Each component also has its own complexities to handle for them to work well together (e.g., To have DIAL working properly in this setting is non-trivial as direct gradient connections can only happen when agents are in communicative states).  As supported by reviewer 4J9r, it is “a thoughtful choice of combination of methods”. Besides, even if the method is considered as simple, we do not think that this should detract from the method’s effectiveness and novelty, and be considered a weakness.
>
> > **Because in the current setting, the communication action directly affects part of the information in obs, does this mean that it is easier to find the communication status by MI? Can such an approach be extended to other scenarios?**
>
> Indeed. The MI term we propose has a direct relationship with communication status. This is further illustrated in our experiments with noisy booth in which we show how noise inversely correlates with MI because it increases the probability of communication failure. So this can indeed be used for communication status. Furthermore, in addition to explicit communication mediums like radio signals and phone lines, our formulation and approach apply to settings without explicit mediums too. For instance, if a robot wants to signal to another one with lights or when satellites need to communicate with each other using lasers, they would need a line of sight to transmit that signal [1] (the channel is light in this case). Naturally, this is spatially constrained and the MI measure can be used to detect whether the transmission is successful or not. One can think of it as another form of ‘influence’ measure, suitable for this more realistic MARL setting.
>
> [1] Lakshmi, K. Shantha, MP Senthil Kumar, and K. V. N. Kavitha. "Inter-satellite laser communication system." In 2008 International Conference on Computer and Communication Engineering, pp. 978-983. IEEE, 2008.
>
>
> > **Clarity: Need to improve**
>
> Can you elaborate more on parts where we can improve our clarity in writing?
>
> > **Reproducibility: Author's open source code is required**
>
> We have added sample code to improve reproducibility.

---

> ### Author Response · Authors · 2022-11-16
> **Has our response addressed your concerns?**
>
> As we are near the end of the discussion period, we would be grateful if the reviewer can confirm whether our response has addressed their concerns, and let us know if any issues remain so we can try to address them before the discussion period ends. Thank you very much!

---

> > ### Comment · Reviewer_Hktd · 2022-11-18
> > **Reply To The Authors**
> >
> > Dear Authors.
> >
> > Thank you for your detailed replies, the current ones solved my confusion. I will consider the comments of other reviewers and then further consider my decision.

---

> > > ### Author Response · Authors · 2022-11-18
> > > **Thank you**
> > >
> > > We are glad that our response has cleared up your confusion. Please do let us know if there are any further concerns.

---

### Official Review · Reviewer_eTc7 · 2022-10-24

**Confidence:** 2
**Correctness:** 3
**Technical Novelty And Significance:** 2
**Empirical Novelty And Significance:** 2
**Recommendation:** 5

**Clarity, Quality, Novelty And Reproducibility:**

The paper is a bit hard to follow. The approach proposed builds on OBL and DIAL, which are described briefly. Readers need to be already familiar with them. The novelty seems limited to the decomposition of the problem in two steps. The use of mutual information is not novel, but is likely novel in he specific context described. Reproducibility is not clear. There is pseudo-code for the method, but I am not sure it will be obvious how to produce code from it. There is no indication of the availability of the software.

**Strength And Weaknesses:**

Strengths: the formalization of the problem to learn how to  communicate is divided in two separate steps, which simplifies the learning process.
Weaknesses: the specific problem of discovery phone booths and learning how to communicate seems artificially complex.  It is not clear the restrictions assumed are realistic, and will make sense in a variety of application domains. The setting of the problem seems to be artificial, designed for the purpose of using existing methods, OBL and DIAL.

**Summary Of The Paper:**

The paper proposes a method for agents to learn how to communicate using free communication channels, named "cheap talk channels". The problem is divided into two separate problems, namely to discover cheap talk and to utilize cheap talk. There is a communication space, separate from the environment action space. Not all states allow communication, so an agent can communicate only if it gets to a state that allows communication. The method proposed is base on maximizing mutual information. Since discovery is separate from utilization, when agents discover a communication channel, they do need to learn how to use the communication channel. To learn how to utilize the communication channel, the agent use DIAL, an existing method to learn efficiently communication protocols. The experimental work covered in the paper is done for a problem where the agents have to discover phone booths before learning how to use them to communicate.

**Summary Of The Review:**

The method proposed seems incremental, built on existing algorithms. The specific example used to illustrate the problem seems to have been artificially created to justify the use of existing algorithms.

---

> ### Author Response · Authors · 2022-11-10
> **Response to Reviewer eTc7**
>
> We appreciate your helpful feedback and comments. We will address your main concerns on the problem setting, clarity, and reproducibility below.
>
> >**Weaknesses: the specific problem of discovery phone booths and learning how to communicate seems artificially complex. It is not clear the restrictions assumed are realistic, and will make sense in a variety of application domains. The setting of the problem seems to be artificial, designed for the purpose of using existing methods, OBL and DIAL.**
>
> We respectfully disagree with the reviewer’s comment that the setting seems artificial, and designed for the proposed method. As supported by reviewer Hktd and reviewer 4j9r, by removing the assumptions that communication channels have to be known apriori and constantly accessible, the problem setting is more realistic than the existing MARL communication setting. Most, if not all, real-world scenarios have spatially constrained communication channels (e.g. inter-satellite laser communication [1]). Under this lens, to communicate successfully, the capability to “learn where (best) to communicate” becomes essential, and our work is the first to consider this more realistic setting in MARL. More importantly, this formulation is a strict generalization of the commonly used setting in emergent communication, and hence, has a broader set of practically relevant applications (please refer to Appendix A for an in-depth justification of the problem setting and an updated introduction in the PDF). On the contrary to assuming more restrictions as mentioned by the reviewer, we are removing unrealistic assumptions on communication channels to have a more realistic setting.
>
> The problem breakdown is not designed for the proposed method. It is a natural breakdown of this more challenging and realistic setting for multi-agent communication. Precisely, when communication is spatially constrained, learning where to communicate has to naturally come before learning to communicate (forming a protocol). Agents can only learn to communicate when they are in communicative states (in other states, communication is ineffective), and this cannot be done effectively unless the agents know where to communicate. Therefore, the breakdown comes naturally from the problem. Our proposed solution subsequently comes from experimentation and understanding of the problem. We show that OBL is a good base algorithm for channel discovery together with MI maximization without forming protocols, and DIAL is an exemplary algorithm to solve the second subproblem in learning a protocol. The proposed method offers the first solution to this challenging setting, based on the natural breakdown of the problem.
>
> [1] Lakshmi, K. Shantha, MP Senthil Kumar, and K. V. N. Kavitha. "Inter-satellite laser communication system." In 2008 International Conference on Computer and Communication Engineering, pp. 978-983. IEEE, 2008.
>
> > **The paper is a bit hard to follow. The approach proposed builds on OBL and DIAL, which are described briefly. Readers need to be already familiar with them. The novelty seems limited to the decomposition of the problem in two steps. The use of mutual information is not novel, but is likely novel in he specific context described. Reproducibility is not clear. There is pseudo-code for the method, but I am not sure it will be obvious how to produce code from it.**
>
> Are there any parts on OBL or DIAL that you would like more clarification or details on? Given that we also have to introduce and formulate this new setting, we distill the algorithmic details of OBL and DIAL while making sure their uses are well justified for the problem. For instance, our description of OBL, though compact, it covers all the necessary elements to understand OBL’s role in this work, in a form similar to the OBL paper. In addition to the pseudocode, Appendix B and C also provide further details on how OBL and DIAL work in our setting. We would be happy to provide any clarifications on OBL and DIAL.
>
> > **There is no indication of the availability of the software.**
>
> We have added sample code in our submission for reproducibility purposes.

---

> ### Author Response · Authors · 2022-11-16
> **Has our response addressed your concerns?**
>
> As the discussion period is coming to an end, we would be grateful if the reviewer can confirm whether our response has addressed their concerns, and let us know if any issues remain so we can try to address them before the discussion period ends. Thanks a lot!

---

> > ### Comment · Reviewer_eTc7 · 2022-11-29
> > **Reply to the Authors**
> >
> > I agree with you on the importance of discovering communication channels and their availability. I still find the specific example of phone boots a bit artificial, but I appreciate your explanation and do not object to it.  I am satisfied that you have made the code available. My enthusiasm for the paper is mitigated by the fact that I have found the paper hard to read, even after reading the extensive Appendices. I am also moderately concerned about the complexity of the innovation since OBL and DIAL are used as major building blocks.

---

> > > ### Author Response · Authors · 2022-12-01
> > > **Reply to Reviewer eTc7**
> > >
> > > Thank you for your response. We are glad that you agree with us on the problem’s importance and appreciate our explanation and code sample. We hope our response has addressed the first weakness you outlined.
> > >
> > > On "the environment is artificial", despite not being an environment that directly emulates a real-world setting, we argue that this remains a crucial first step to provide a benchmarking tool for researchers to investigate this problem setting. Even if it is relatively simple, it still has all the necessary properties to evaluate existing algorithms with an appropriate level of difficulty (as we show how existing baselines fail in this environment). Together with our novelty in problem formulation and method, we believe the work serves as an important contribution to the field, encouraging more investigations in more realistic MARL settings.
> > >
> > > On the paper being hard to read, we will continue to identify places to improve the paper’s clarity on some of the ideas. If you have any specific parts/paragraphs of the paper that you find requires further elaboration or clarification, please let us know. We would greatly appreciate the helpful advice and make sure they are addressed in the next iteration of the paper.
> > >
> > > Regarding complexity, we understand that the method has a nontrivial level of complexity. But we respectfully argue that the level of complexity is well-motivated and sufficiently necessary. The combination of methods is a meticulous and thoughtful decision as supported by reviewer 4J9r, based on our understanding of the problem setting. For instance, OBL is a careful design choice given our understanding that no protocols should be formed during cheap talk discovery. This is further supported by our ablation studies that demonstrate the importance of each proposed component, aligning with our motivations (e.g., we show that replacing OBL with IQL during cheap talk discovery significantly worsens performance). Hence, we strongly believe that the method’s level of complexity is sufficiently necessary and natural (as we elaborated in our last response).

---

### Official Review · Reviewer_3EzS · 2022-10-30

**Confidence:** 4
**Correctness:** 3
**Technical Novelty And Significance:** 3
**Empirical Novelty And Significance:** 2
**Recommendation:** 6

**Clarity, Quality, Novelty And Reproducibility:**

Clarity
Problem setting and definitions can be made clearer and can help in making a much stronger case for the paper's main arguments.

Quality
Current set of chosen baselines makes it difficult to assess the quality of the proposed model. More immediate competitors should be tested under more varied environments. Comments on PMI computation should accompany evaluation results. On a different note, the heatmap should probably be explained in more detail since it may not be the most conventional visual aid in related literature.

Originality
Mutual information maximization is not a novel training objective. It is difficult to evaluate the novelty of A_comm since it can be better clarified. OBL is also an independent previous work. Put together, groundbreaking novelty might not be the paper's appeal, but the structured combination of PMI and OBL does work well in harshly limited settings.

**Strength And Weaknesses:**

Paper addresses a well-defined niche by relaxing a set of common assumptions found in some communication-based MARL methods. The tested evaluation environment is intuitive. Claims made with an intuitive approach (MI maximization) are shown to function correctly in the empirical evaluation.

A number of factors can make the paper much stronger: problem positioning, clarified definition (especially action and channel), varied environments, better justified baseline choice, and comments on computation overhead.

While the problem addressed is clearly an interesting setting, real-world settings might question its practicality. If the communication channels are not even known a priori, could there really be multiple agents who happen to have access to the same channels and also coincidentally need to cooperate? Consider a swarm of drones. If one common administrator is operating all of those drones as in a search-and-rescue operation, it is almost always the case that their communication channels are known pre-dispatch. On the other hand, if different operators are managing each of those drones as in individual drone racing competitions, then it may well be the case that there simply is no need for cooperative/coordinated flight management. It feels as though the removal of assumptions inadvertently led to a less realistic problem setting, so a better problem positioning could more effectively highlight the significance of the proposed solution to the problem.

Communicative actions and channels could be defined more clearly. If the term "channels" are correctly aligned with the same term in the radio communications sense, then communicative actions being "hint up" or "hint down" can be a misinterpretation of communication. Instead, should the concepts of channels remain faithful to radio communications literature, then the problem of which channels to discover should also pertain to the communication channels, such as which bandwidths. The current formulation of A_comm is probably more aptly described as message encoding since the chosen action answers the question of "which message should be sent". However, if this indeed is the case, then CTDUL's A_comm would simply be hard-coded actions.

It would be a valuable addition to the paper to investigate CTDUL's performance in other environments, such as SMAC or MAMuJoCo.

It would also be an insightful curation of CTDUL's positioning to include SchedNet (ICLR 2019), which addressed medium contention and medium access constraints in MARL, and DIAYN (2018), which is an information-theoretic approach to RL exploration and then identify CTDUL's strengths, differences, and delimitation.

Please also elaborate on the cost of computing pairwise mutual information and how this affects the empirical performance of CTDUL.

**Summary Of The Paper:**

Authors design and implement an algorithm that operates under fewer assumptions than some communication-based MARL works. Agents learn to discover which environment actions and communicative actions to take, so as to maximize the mutual information (MI) with respect to observation at the receiver's end. The proposed CTDUL outperforms several baselines in the phone booth maze environment.

**Summary Of The Review:**

Proposed method works well for a specific setting, whose characteristics can be better marketed as more realistic. Some definitions need clarification, and related literature can benefit from a closer investigation into the papers suggested above.

---

> ### Author Response · Authors · 2022-11-10
> **Response to Reviewer 3EzS (3/3)**
>
> > **It would be a valuable addition to the paper to investigate CTDUL's performance in other environments, such as SMAC or MAMuJoCo.**
>
>
> As the current setting already poses significant difficulty to the baseline algorithms, we kept our analysis within this setting to avoid unnecessary complexity and to allow a more in-depth analysis of the novel problem setting and the proposed method. We agree that it would be valuable to extend our work to more environments. But we believe this work serves as a good starting point to encourage more investigations in more realistic settings for MARL communication, by imposing the realistic need to learn where to communicate. Existing environments like SMAC and MAMuJoCo, do not support spatially constrained communication immediately and have no communication obstacles. This means our proposed method would become equivalent to DIAL. Hence, evaluating our method in these environments would not yield additional insights. Nonetheless, we aim to create more complex environments with spatially constrained communication for future work.
>
> > **It would also be an insightful curation of CTDUL's positioning to include SchedNet (ICLR 2019), which addressed medium contention and medium access constraints in MARL, and DIAYN (2018), which is an information-theoretic approach to RL exploration and then identify CTDUL's strengths, differences, and delimitation**
>
> Thank you for the baseline suggestions. For SchedNet, we think their setting is orthogonal to ours. They are considering a shared medium with limited bandwidth (like a phone line with multiple users at the same time), which is different from our work’s focus - spatially constrained communication. But it would be interesting to have all of these constraints together to have a more realistic multi-agent communication problem, which we leave as future work. For DIAYN, it is a single-agent RL method to discover options and does not directly apply to discovering communication channels. We believe exploring methods like DIAYN is out of scope and serves better as a separate study. We use several more relevant baselines and SOTA methods in MARL as a comparison. We show how existing independent learning methods (IQL) and methods with centralized components (e.g. QMIX) converge to trivial and suboptimal solutions that fail to utilize the communication booths. We also compare with social Influence - SI, a recent method using mutual information maximization, which is the closest to our method and is a variation of influence-based exploration methods. Hence, in the context of MARL, We think our experiments are sufficient to evaluate how existing methods perform compared to our method as we have considered baselines with centralized components and MARL-based exploration.
>
> > **Please also elaborate on the cost of computing pairwise mutual information and how this affects the empirical performance of CTDUL.**
>
> The current implementation requires 1-step rollouts of an agent for each action (both A_env and A_comm) to compute the PMI term. If we have very large A_env and/or A_comm, we can simply perform action sampling for each action subspace. This would provide an approximation of PMI. It would be as effective as the precise term in discovering communication channels because what the agent needs are some rollouts in which communication is successful. Hence, computing costs should not be a concern.
>
>
> > **On a different note, the heatmap should probably be explained in more detail since it may not be the most conventional visual aid in related literature.**
>
> Please let us know if further clarifications are needed on these points as we have addressed them above. We have also clarified the description and caption of the heatmaps in the PDF to make it clearer.
>
> > **Mutual information maximization is not a novel training objective. It is difficult to evaluate the novelty of A_comm since it can be better clarified. OBL is also an independent previous work. Put together, groundbreaking novelty might not be the paper's appeal, but the structured combination of PMI and OBL does work well in harshly limited settings.**
>
> Just to reiterate our contributions and novelty, to the best of our knowledge, this is the first work to address and tackle this more realistic problem setting (i.e., learning where (best) to communicate). This is in agreement with other reviewers’ views too with respect to the setting being novel and uninvestigated in the current literature. We believe the setting and method have significant importance to the field for two reasons. Firstly, as elaborated above, the work has direct relevance to real-world applications using MARL. Secondly, it takes a substantial step toward more scalable and realistic approaches to learning to communicate.

---

> ### Author Response · Authors · 2022-11-10
> **Response to Reviewer 3EzS (2/3)**
>
> > **It feels as though the removal of assumptions inadvertently led to a less realistic problem setting, so a better problem positioning could more effectively highlight the significance of the proposed solution to the problem.**
>
> We respectfully disagree that our assumption removal leads to a less realistic setting, in which reviewers Hktd and 4J9r share the same position. To clarify, channels here are the mediums for which signals or messages can be transmitted between agents. Intuitive examples would be a radio link and a cell phone signal.  Even in communication without an explicit/conventional medium, say if a robot wants to signal to another one with lights or when satellites need to communicate with each other using lasers, they would need a line of sight to transmit that signal (in this case, the channel is light). More often than not, many of these mediums are spatially constrained in real-world scenarios which is the main motivation of this work. [1] shows how important positioning is for inter-satellite laser communication systems to obtain a line of sight. When satellites become autonomous, the ability to position themselves for better communication would be needed.  Under this lens, the problem setting is realistic and very common and can be applied to any setting that requires message transmission between agents. Even if agents have access to known communication channels, given that real-world communication is spatially constrained in many cases, there is still the need to learn where best to communicate. Our work generalizes the MARL communication problem to cover this more realistic need.  Please refer to Appendix A for a more in-depth discussion. We have also modified the introduction to better motivate the work with the satellite example in the PDF.
>
> [1] Lakshmi, K. Shantha, MP Senthil Kumar, and K. V. N. Kavitha. "Inter-satellite laser communication system." In 2008 International Conference on Computer and Communication Engineering, pp. 978-983. IEEE, 2008.
>
> > **Communicative actions and channels could be defined more clearly. If the term "channels" are correctly aligned with the same term in the radio communications sense, then communicative actions being "hint up" or "hint down" can be a misinterpretation of communication. Instead, should the concepts of channels remain faithful to radio communications literature, then the problem of which channels to discover should also pertain to the communication channels, such as which bandwidths. The current formulation of A_comm is probably more aptly described as message encoding since the chosen action answers the question of "which message should be sent". However, if this indeed is the case, then CTDUL's A_comm would simply be hard-coded actions.**
>
> Our formulation follows the standard interpretation in MARL communication without explicit consideration of radio communications. Similar to what the reviewer described, we consider channels as the medium in which messages can be transmitted. They could vary in bandwidth and transmission reliability (for the scope of this work, we modeled it as noise). A_comm indeed can be interpreted as message encoding so the agent can learn what to send. These discrete messages are formulated similarly to what has been used in previous works (e.g. DIAL). The interpretation also aligns with the MARL communication literature [1].  Hence, A_comm should be viewed as discrete messages, not hard-coded actions. To reiterate, the key contribution of this work is to introduce a more realistic setting for communication channels in MARL by having them spatially constrained with varying availability. Hence, the agent has the extra need to learn “where (best) to communicate” before learning “what to communicate” (i.e. using the message encoding).
>
> [1] Angeliki Lazaridou and Marco Baroni. Emergent Multi-Agent Communication in the Deep Learning Era, July 2020.

---

> ### Author Response · Authors · 2022-11-10
> **Response to Reviewer 3EzS (1/3)**
>
> We thank you for your valuable feedback and helpful comments. We address your questions and concerns below.
>
> > **While the problem addressed is clearly an interesting setting, real-world settings might question its practicality. If the communication channels are not even known a priori, could there really be multiple agents who happen to have access to the same channels and also coincidentally need to cooperate? Consider a swarm of drones. If one common administrator is operating all of those drones as in a search-and-rescue operation, it is almost always the case that their communication channels are known pre-dispatch.**
>
> Under our formulation, the drones would already know that they can send different signals through, for example, a radio channel. Similarly, in the phone booth environment, agents are equipped to send discrete signals (only effective in certain states). In remote areas where signal strength could vary, even though the drones are equipped with these communication actions, they would still need to learn "where best to communicate" by positioning themselves optimally. We use phone booths and noise to abstract this problem for the sake of experimentation. But the formulation straightforwardly extends to the drone example. S_comm might be a lot bigger but not every state in S_comm is as effective in reality.
>
> > **On the other hand, if different operators are managing each of those drones as in individual drone racing competitions, then it may well be the case that there simply is no need for cooperative/coordinated flight management.**
>
> Purely competitive scenarios would indeed lessen the need to communicate. But real-world settings are rarely purely competitive and are mostly mixed. In mixed scenarios, it has been shown, e.g., [1], that communication could still be beneficial. Overall, whenever communication is needed (which we believe to be very common in multi-agent systems), our formulation assumes a more realistic set of assumptions than existing works.
>
> [1] Singh, Amanpreet, Tushar Jain, and Sainbayar Sukhbaatar. "Learning when to communicate at scale in multiagent cooperative and competitive tasks." arXiv preprint arXiv:1812.09755 (2018).

---

> ### Author Response · Authors · 2022-11-16
> **Has our response addressed your concerns?**
>
> As we are near the end of the discussion period, we would be grateful if the reviewer can confirm whether our response has addressed their concerns, and let us know if any issues remain so we can try to address them before the discussion period ends. Thanks a lot!

---

### Author Response · Authors · 2022-11-10
**Paper Update**

We have updated the paper with respect to reviewer's comments and put all our changes in red.

---

### Decision · Program_Chairs · 2023-01-20

**Decision:**

Accept: poster

**Justification For Why Not Higher Score:**

The paper studies an interesting problem for the multi-agent reinforcement learning problem. Although all the reviewers agree the paper can be accepted, they also raised valid concerns regarding the readability and the practicality of the examples studied in the paper.

**Justification For Why Not Lower Score:**

The paper studies an interesting problem for the multi-agent reinforcement learning problem (with communication issues). Both the novelty and technique contributions are significant enough for the paper to be accepted.

**Metareview: Summary, Strengths And Weaknesses:**

Summary:
This paper establishes an interesting scenario for the multiagent RL that there exist states where agents can communicate with each other. These communication channels are called the "cheap-talk" channels. This paper introduces a novel conceptual framework for the "cheap-talk discovery" and "cheap-talk utilization" steps and develops a new algorithm based on mutual information maximization that outperforms existing algorithms. This paper also releases a novel benchmark suite to stimulate future research.

Strengths:
- A simple and effective framework for the communication problem in the multi-agent setting.
- The authors provide an experimental environment benchmark.
- Experiments show that their approach has significant advantages.
- Code available

Weakness:
- The scenario studied in this paper might be less practical (or even artificial).
- The paper may be hard to read.

During the rebuttal phase, the authors provided a number of clarifications and also made the code available. The authors' responses address a number of the concerns of the reviewers, and all of them agree the paper can be accepted.

**Note From Pc:**

if the above contains the word "oral" or "spotlight" please see: "oral" presentation means -> notable-top-5% and "spotlight" means -> notable-top-25%. As stated in our emails, we are disassociating presentation type from AC recommendations